# 27-Hydroxycholesterol-Induced Dysregulation of Cholesterol Metabolism Impairs Learning and Memory Ability in ApoE ε4 Transgenic Mice

**DOI:** 10.3390/ijms231911639

**Published:** 2022-10-01

**Authors:** Yushan Wang, Ling Hao, Tao Wang, Wen Liu, Lijing Wang, Mengwei Ju, Wenjing Feng, Rong Xiao

**Affiliations:** School of Public Health, Beijing Key Laboratory of Environmental Toxicology, Capital Medical University, No. 10 Xitoutiao, You An Men Wai, Beijing 100069, China

**Keywords:** 27-hydroxycholesterol, cholesterol metabolism, homeostasis, learning and memory impairment, transgenic mice

## Abstract

Dysregulated brain cholesterol metabolism is one of the characteristics of Alzheimer’s disease (AD). 27-Hydroxycholesterol (27-OHC) is a cholesterol metabolite that plays an essential role in regulating cholesterol metabolism and it is suggested that it contributes to AD-related cognitive deficits. However, the link between 27-OHC and cholesterol homeostasis, and how this relationship relates to AD pathogenesis, remain elusive. Here, 12-month-old ApoE ε4 transgenic mice were injected with saline, 27-OHC, 27-OHC synthetase inhibitor (anastrozole, ANS), and 27-OHC+ANS for 21 consecutive days. C57BL/6J mice injected with saline were used as wild-type controls. The indicators of cholesterol metabolism, synaptic structure, amyloid β 1-42 (Aβ1-42), and learning and memory abilities were measured. Compared with the wild-type mice, ApoE ε4 mice had poor memory and dysregulated cholesterol metabolism. Additionally, damaged brain tissue and synaptic structure, cognitive decline, and higher Aβ1-42 levels were observed in the 27-OHC group. Moreover, cholesterol transport proteins such as ATP-binding cassette transporter A1 (ABCA1), apolipoprotein E (ApoE), low-density lipoprotein receptor (LDLR), and low-density lipoprotein receptor-related protein1 (LRP1) were up-regulated in the cortex after the 27-OHC treatment. The levels of cholesterol metabolism-related indicators in the hippocampus were not consistent with those in the cortex. Additionally, higher serum apolipoprotein A1 (ApoA1) levels and lower serum ApoE levels were observed in the 27-OHC group. Notably, ANS partially reversed the effects of 27-OHC. In conclusion, the altered cholesterol metabolism induced by 27-OHC was involved in Aβ1-42 deposition and abnormalities in both the brain tissue and synaptic structure, ultimately leading to memory loss in the ApoE ε4 transgenic mice.

## 1. Introduction

Alzheimer’s disease (AD) is a multifactorial neurodegenerative disorder characterized by impaired cognitive function and a combination of neuropsychiatric and motor symptoms [1]. Neuropathologically, it is characterized by the deposition of amyloid plaques and neurofibrillary tangles [2]. However, the pathogenesis of this disease remains poorly understood.

The brain is rich in cholesterol, with almost 25% of the total amount in the body [3]. Cholesterol metabolism is a complex and highly regulated process comprising cholesterol biosynthesis, transport, esterification, and efflux [4]. Cholesterol metabolism is identified as a modulator of AD through its involvement in regulating neurotransmission, synapse formation, and synaptic plasticity [5]. In a genome-wide association study, most of the genes in late-onset AD were related to cholesterol metabolism pathways [6]. Animal studies have revealed that a high cholesterol level could cause learning and memory decline [7,8]. However, the precise mechanism by which alterations in cholesterol metabolism contribute to AD pathogenesis is not fully understood.

Brain cholesterol is primarily synthesized in situ by astrocytes. It is almost completely isolated from peripheral cholesterol by the blood–brain barrier (BBB) [9]. The sterol 27-hydroxylase (CYP27A1) can catalyze the hydroxylation of cholesterol to form 27-OHC mainly in the liver, which is initially released into the blood. 27-OHC is an abundant oxysterol in the blood with an ability to cross the BBB and it plays an essential role in AD pathogenesis [10,11,12]. An increasing amount of evidence has demonstrated that the plasma level of 27-OHC in patients with mild cognitive impairment and AD is higher than average [13,14]. In addition, our previous studies have shown that 27-OHC contributes to cognitive decline in APP/PS1 and C57BL/6J mice [15], accelerates the accumulation of Aβ both in vitro and in vivo [15,16], damages synaptic structures, and results in synaptic dysfunction in primary hippocampal neurons [17]. Moreover, 27-OHC plays an essential role in cholesterol metabolism. Our previous works have indicated that 27-OHC could inhibit cholesterol synthesis in C6 glioma cells [18] and facilitate the transport of cholesterol from astrocytes to neurons [16]. However, the precise mechanism by which 27-OHC regulates cholesterol transport, esterification, and efflux, as well as its role in the pathogenesis of AD, remain unclear. 

Apolipoprotein E (ApoE), an apolipoprotein encoded by the *APOE* gene, plays a pivotal role in cholesterol transport [19,20]. ApoE has been reported to modulate Aβ levels and AD pathogenesis [19]. The ApoE ε4 allele is a major genetic risk factor for AD [21]. Multiple clinical studies have demonstrated that APOE ε4-carriers are disproportionately susceptible to the onset of—and neuronal deterioration caused by—AD [22,23,24,25]. *APOE* is also strongly associated with cognitive impairment, Aβ aggregation [19,20], and synaptic deficit [26], as well as the breakdown of BBB [27,28]. These findings suggest that ApoE has an obvious effect on cholesterol metabolism and may induce multiple pathological changes in AD by regulating cholesterol transport.

Herein, we used ApoE ε4 transgenic mice to explore the possible mechanisms of how excessive 27-OHC levels induced cholesterol metabolism dysregulation and neurotoxicity. We propose that excessive 27-OHC levels could disturb cholesterol transport, esterification, and efflux, ultimately leading to abnormalities in the both brain tissue and synaptic structure, Aβ1-42 deposits, and a decline in the learning and memory abilities in ApoE ε4 transgenic mice. 

## 2. Results

### 2.1. Bodyweight and Organ Coefficient of Mice

The body weight and the organ coefficient (organ weight/body weight) of mice were measured to evaluate the effects of the treatments on growth and organ weight. As shown in Figure 1A, there was no significant change in the bodyweight of the mice before and after treatment. The heart (F = 18.312; *p* < 0.001) and kidney (F = 10.441; *p* < 0.001) coefficients were lower in the Model group compared with the WT Control group (Figure 1B,D). In addition, compared with the Model group, the liver coefficient in the WT Control group (*p* = 0.035), 27-OHC group (*p* = 0.048), and 27-OHC+ANS group (*p* = 0.015) were significantly lower (Figure 1C). The liver coefficient was higher in the ANS group than in the 27-OHC+ANS group (*p* = 0.02) (Figure 1C). The spleen coefficient did not differ between groups (Figure 1E).

### 2.2. Morris Water Maze

The water maze test was performed to detect the effect of 27-OHC and its synthase inhibitor, anastrozole (ANS), on the learning and memory abilities of mice. As shown in Figure 2A, in the training phase, the escape latency was significantly lower on Day 5 compared with Day 1 (F = 116.854; *p* < 0.001), with marked differences in the escape latency among the mice on Day 5 (F = 14.423; *p* < 0.001). The escape latency in the Model group (*p* < 0.001) and 27-OHC+ANS group (*p* = 0.39) was shorter than that in the 27-OHC group (Figure 2A), suggesting that the 27-OHC could impair memory and learning functions and that ANS could partially reverse these effects. In the test phase, the number of target platform crossings in the Model group was significantly lower compared with that in the WT Control group (Figure 2C). The time in the target quadrant was markedly lower in the 27-OHC-treated group. In contrast, the mean distance to the platform of the mice was higher in the 27-OHC group and 27-OHC+ANS group compared with the Model group (Figure 2E,G), which demonstrated that the 27-OHC treatment impaired memory and learning functions. The distance of the target quadrant in the ANS group was higher than that in the 27-OHC group (Figure 2F). No significant difference was observed with respect to the average speed (*p* > 0.05) (Figure 2D).

### 2.3. 27-OHC Damages Brain Tissue Structure and Nerve Synapses in Mice

Nissl staining was performed to assess the effect of the 27-OHC treatment on hippocampal tissue structure, as shown in Figure 3A. The hippocampal structure of the WT mice was clear, and the vertebral body layer cells were intact, layered, and arranged tightly. In the Model group, 27-OHC group, ANS group, and 27-OHC+ANS group, some cells had incomplete morphology, indistinct nucleoli, and fewer Nissl bodies, with vacuole-like changes in specific cells.

To further assess the effect of 27-OHC on the plasticity of nerve synapses, we observed the microstructures of the synapses using a transmission electron microscope. As shown in Figure 3B,C, after the 27-OHC treatment, the number of synaptic vesicles (*p* = 0.01) and the area of the postsynaptic membrane (*p* < 0.001) were significantly reduced in the hippocampal CA1 region.

Synapse-related proteins, such as growth-associated protein 43 (GAP43), postsynaptic density protein 95 (PSD95), activity-regulated cytoskeleton-associated protein (Arc), synaptophysin (SYN), synaptosome-associated protein 25 (SNAP-25), and microtubule-associated protein 2 (MAP2), were detected by western blot. In the hippocampus, compared with the Model group, the GAP43 in the 27-OHC group (*p* = 0.012) was down-regulated (Figure 4A), whereas GAP43 (*p* = 0.011) and SNAP-25 (*p* = 0.002) levels in the ANS group were significantly up-regulated (Figure 4A,E). The GAP43 (*p* < 0.001) and SNAP-25 (*p* < 0.001) levels in the 27-OHC+ANS group were significantly down-regulated compared with the ANS group (Figure 4A,E). In the cortex area, compared with the Model group, the GAP43 levels in the WT Control group (*p* = 0.011), 27-OHC group (*p* < 0.001), ANS group (*p* = 0.006), and 27-OHC+ANS group (*p* < 0.001) were down-regulated, whereas GAP43 in the ANS group was significantly up-regulated compared with the 27-OHC group (Figure 4G). In addition, the expression of MAP2 in the 27-OHC+ANS group was lower than that in the ANS group (Figure 4L).

### 2.4. 27-OHC Increases Aβ Burden in Hippocampus and Cortex of ApoE4 Mice

Immunohistochemical analysis was used to detect the Aβ burden in the brain. The results showed fewer Aβ1-42-positive areas in each group. Nevertheless, the proportion of Aβ1-42 in the DG area of the hippocampus in the Model group was significantly higher than that in the WT Control group (*p* < 0.001). The Aβ1-42 levels in the cortical area (*p* < 0.001) and hippocampal CA1 area (*p* < 0.001) in the 27-OHC group were significantly higher than those in the Model group. An opposite result was obtained in the hippocampal DG area. In all four regions, the Aβ1-42-positive areas in the 27-OHC+ANS group were fewer than those in the 27-OHC group, but not statistically different in the hippocampal CA1 area. (Figure 5).

### 2.5. Serum Cholesterol Levels

As shown in Figure 6, the Model group had higher total cholesterol (TC) and lower HDL-C/TC levels than the WT Control group. Compared with the Model group, the level of TC was lower in the 27-OHC group (*p* = 0.008) and 27-OHC+ANS group (*p* = 0.015) (Figure 6A). The level of LDL-C was decreased in the 27-OHC group (*p* = 0.003), ANS group (*p* = 0.035), and 27-OHC+ANS group (*p* = 0.001) (Figure 6C).

### 2.6. 27-OHC Affects Brain Cholesterol Transport

To detect the effect of 27-OHC on brain cholesterol transport, the expression of cholesterol transport-related proteins—including ATP-binding cassette transporter A1 (ABCA1), ATP-binding cassette transporter G1 (ABCG1), ATP-binding cassette transporter G4 (ABCG4) and apolipoprotein E (ApoE), low-density lipoprotein receptor (LDLR), low-density lipoprotein receptor-related protein1 (LRP1), and scavenger receptor class B type 1 (SR-BI)—was analyzed in the hippocampus and cortex. It was shown that the ApoE (*p* < 0.001) protein level was significantly higher in the Model group compared with the WT Control group in the hippocampus (Figure 7M), whereas the levels of ABCA1 (*p* = 0.008), LDLR (*p* < 0.001), and LRP1 (*p* = 0.046) were lower (Figure 7A,H,I). The levels of ABCA1 (*p* = 0.002), LDLR (*p* = 0.011), and ApoE (*p* < 0.001) were markedly lower in the 27-OHC group compared with the Model group. A similar result was also found when comparing the 27-OHC+ANS group with the ANS group (Figure 7A,E,M). In addition, the level of LDLR in the 27-OHC+ANS treatment was higher than in the 27-OHC group (*p* = 0.022) (Figure 7E). In the cortex, significantly higher levels of ApoE (*p* < 0.001) and LDLR (*p* = 0.019) protein and lower ABCA1 (*p* < 0.001) and LRP1 (*p* = 0.013) levels were found in the Model group compared with the WT Control group (Figure 7G,K,L,N). The levels of ABCA1 (*p* = 0.001), LDLR (*p* = 0.011), LRP1 (*p* = 0.837) and ApoE (*p* = 0.016) protein were significantly up-regulated in the 27-OHC group (Figure 7G,K,L,N). 

In addition, the serum ApoE level in the Model group (*p* < 0.001) was significantly higher than that in the WT Control group (*p* < 0.001) and markedly lower in the 27-OHC group than in the Model group (*p* = 0.007) (Figure 7O). In contrast, the level of apolipoprotein A1 (ApoA1) showed opposite trends. Moreover, ANS could only slightly decrease the level of ApoA1 (Figure 7P).

### 2.7. 27-OHC Affects Brain Cholesterol Conversion

Excess brain cholesterol is converted into cholesterol esters and oxysterols. Therefore, we detected the expression levels of cholesterol esterase and oxysterol synthase. As shown in Figure 8, in the hippocampus, the acetyl-coenzyme A acetyltransferase 1 (ACAT1) protein was significantly higher in the 27-OHC group (*p* = 0.001) and lower in the ANS group (*p* < 0.001) and 27-OHC+ANS group (*p* < 0.001) compared with the Model group (Figure 8A). However, no significant change was observed in the cortex (*p* > 0.05) (Figure 8C). In addition, the expression level of cytochrome P450 family 46 subfamily A polypeptide 1 (CYP46A1) in the Model group was higher than that in the WT Control group (*p* = 0.04) in the hippocampus (Figure 8B). In contrast, in the 27-OHC+ANS group, cortical CYP46A1 protein expression was slightly higher than that in the 27-OHC group (*p* = 0.043) (Figure 8D).

## 3. Discussion

Cholesterol metabolism has been identified as a modulator of AD that is involved in maintaining synaptic plasticity, neural development, and brain function [5]. Evidence suggests that 27-OHC plays an essential role in AD [11,12,29,30]. Moreover, our previous studies have shown that 27-OHC could inhibit cholesterol synthesis in C6 glioma cells [18] and facilitate the transport of cholesterol from astrocytes to neurons [16]. This study used the exogenous 27-OHC and CYP27A1 inhibitor, ANS, to treat ApoE ε4 transgenic mice. We found that excessive 27-OHC levels could induce morphologic damage in both brain tissue and nerve synapses and increase the burden of Aβ1-42 in brain tissue, ultimately leading to learning and memory impairment. The biological mechanism of 27-OHC-induced AD pathogenesis may involve the cholesterol metabolism processes in peripheral and brain tissues. In addition, 27-OHC synthetase inhibitor could partially alleviate the effect of 27-OHC.

This study first explored the link between 27-OHC and cognitive impairment. The water maze test found that ApoE ε4 mice have a longer escape latency and fewer target crossings than WT mice, supporting the theory that *APOE4* is an AD risk gene [31,32]. In addition, the 27-OHC group had a longer escape latency, mean distance to the platform, and a lower time in the target quadrant, indicating that 27-OHC plays a negative regulatory role in cognitive function [33]. Moreover, the escape latency and time in the target quadrant in the 27-OHC+ANS group were shorter than in the 27-OHC group, suggesting that ANS could inhibit the toxic effect of 27-OHC and ameliorate the decline in learning and memory abilities [33,34]. 

Synaptic loss occurs early and is strongly correlated with cognitive impairment in AD [35]. A transmission electron microscope was used to observe changes in the brain microstructures more closely. The hippocampus is crucial for learning and memory functions [36]. In this study, we observed the synaptic microstructures of nerve synapses in the hippocampus’ CA1 area, which is considered an important morphological structure for processing long-term memory [37]. The number of synaptic vesicles was significantly lower in the Model group compared with the WT Control group, whereas no changes were observed in the postsynaptic membrane area. Moreover, 27-OHC reduced the area of the postsynaptic membrane and the number of synaptic vesicles; additionally, CYP27A1 inhibitors (ANS) alleviated the toxic effect of 27-OHC on nerve synapses to a certain extent. These results are consistent with those of Paula’s study and our previous in vitro experiment [17,38]. In addition, GAP43 was significantly down-regulated in the 27-OHC group in both the hippocampus and cortex, but slightly up-regulated following the ANS treatment. Similar results were observed for MAP2 expression in the cortex and SNAP-25 in the hippocampus. These results are consistent with the finding by Valencia et al. [39] that high concentrations of 27-OHC could reduce the dendritic density of pyramidal neurons in the hippocampus CA1 area and PSD95 expression. This is also consistent with our previous finding that 27-OHC could reduce the number and length of synapses in primary hippocampal neurons, damage their synaptic microstructures, and down-regulate the expression of synaptic proteins [17].

Amyloid β (Aβ) is a key pathogenic factor in AD. High levels of cholesterol in brain cell membranes can make the lipid rafts more stable and promote Aβ accumulation in the brain parenchyma [20,40]. The present study found that 27-OHC increased brain cholesterol levels and Aβ1-42 deposition in the cortical area and hippocampal CA1 area, confirming the hypothesis that high levels of 27-OHC could promote AD-like lesions by regulating Aβ1-42 in ApoE ε4 mice. However, 27-OHC had an opposite effect on Aβ1-42 deposition in the hippocampal DG area. We speculated that this divergence could have occurred because the deposition of Aβ1-42 in the hippocampal DG region was less than that in the CA1 and CA3 regions, and Aβ1-42 deposit may preferentially accumulate in CA1 and CA3 regions [41].

Cholesterol plays a pivotal role in cognitive function. Our previous study showed that dietary cholesterol could increase the plasma level of 27-OHC and impair rats’ learning and memory abilities [7]. The present study found that the serum TC level in the Model group was significantly higher than that in WT Control group, which is consistent with the results of the studies by Dankner [42] and Wang [43]. However, compared with the Model group, the 27-OHC group had lower serum TC and LDL-C levels. This might be due to the availability of LDLR, which could trap ApoE4 and is involved in the clearance of LDL-C in plasma [44,45]. In this study, we found that plasma ApoE levels were reduced after 27-OHC treatment, potentially increasing the availability of LDLR for the rapid clearance of plasma LDL-C. However, LDL-C/TC and LDL-C/HDL-C demonstrated a weak rising trend in the 27-OHC group, confirming our hypothesis that 27-OHC could cause dysregulated cholesterol metabolism.

Brain cholesterol is mainly synthesized by astrocytes and can be transported between different cells by combining it with ApoE [46]. ABCA1 and ABCG1 in astrocytes mediate the binding of cholesterol to ApoE. These lipoproteins can be used by the neuronal lipoprotein receptors LDLR, LRP1, and SR-B1 and be endocytosed into cells; then, after a series of hydrolysis reactions, free cholesterol is released for use by neurons [47]. In the present study, the ApoE ε4 mice had higher ApoE and lower ABCA1 and LRP1 protein expression levels in both the hippocampus and cortex than the WT mice, suggesting that the ApoE ε4 genotype affects the expression of ABCA1 and LRP1 [48,49]. Additionally, 27-OHC up-regulated ABCA1, LDLR, LRP1 and ApoE in the cortex, which was consistent with our earlier finding that 27-OHC could increase ABCA1 protein expression in C6 cells in a dose-dependent manner [16]. This finding is compatible with the results detailing how 27-OHC is a liver X receptor (LXR) ligand that could upregulate the LXR-responsive gene *ABCA1* [50]. However, opposite results were obtained in the hippocampus, which probably reflects its distinct protective mechanisms or the involvement of a region-specific pathology in the brain [51]. It was also demonstrated that ABCA1, LDLR, and LRP1 are involved in the removal of Aβ [52,53,54], which has a much higher deposition in the cortex than in the hippocampus. Thus, the higher the Aβ in the cortex, the more ABCA1, LDLR, and LRP1 accumulate in the brain to eliminate it [55]. In the present study, the serum ApoE level was higher in the Model group compared with the WT Control group, whereas the ApoA1 level was lower. Additionally, 27-OHC caused a decrease in the serum ApoE levels and an increase in ApoA1 levels. Both ApoE and ApoA1 participate in peripheral cholesterol transport. The increase in the ApoA1 level could represent a compensatory response after the loss of ApoE-mediated cholesterol transport. Moreover, the liver is the main site for ApoE synthesis, which is then secreted into the blood [56,57].This research found that the organ coefficient of the liver was lower after the 27-OHC treatment. Therefore, excessive 27-OHC levels might impair liver function and reduce the serum ApoE level. Moreover, the brain ApoE level was higher in the Model group than that in the WT Control group. 

Therefore, this study confirmed that the ApoE ε4 genotype regulates cholesterol levels and that 27-OHC may affect brain tissue cholesterol transport through ABCA1-ApoE-LDLR/LRP1 proteins. However, 27-OHC has different effects on the cholesterol endocytic receptors LDLR and LRP1 in various brain regions.

When the level of cholesterol is elevated in brain tissue, cholesterol acyltransferase1 (ACAT1) converts free cholesterol into cholesterol esters. Moreover, the conversion of cholesterol to 24S-hydroxycholesterol (24S-OHC) by CYP46A1 is also a critical step in brain cholesterol elimination. In this study, 27-OHC significantly increased ACAT1 protein expression in the hippocampus, suggesting that 27-OHC results in increased cholesterol esters. This is consistent with the finding that three transgenic familial AD mice had increased cholesterol esters [58]. ACAT1 inhibition reduces cerebral Aβ [59]. These findings support our hypothesis that 27-OHC regulates cognitive function by altering cholesterol esterification. Expressed mainly in the brain, CYP46A1 is a major regulator of brain cholesterol elimination [60]. A recent cohort experiment confirmed that the level of 24S-OHC and the 24S-OHC/27-OHC ratio in the cerebrospinal fluid of AD patients was higher, which predicted an earlier cognitive decline [61]. In addition, our in vitro experiment confirmed that 27-OHC could facilitate CYP46A1 translocation from the endoplasmic reticulum to the cell membrane [16]. In the present study, the Model group showed a significant increase in CYP46A1 protein expression in the hippocampus compared with the WT Control group. In addition, the CYP46A1 level was higher in the 27-OHC group than that in the ANS group. This may largely be explained by the cholesterol accumulation in the Model group and 27-OHC group. In addition, the CYP46A1 expression level was significantly higher in the 27-OHC+ANS group compared with the 27-OHC group in the cortex, suggesting that the CYP27A1 inhibitor (ANS) could up-regulate CYP46A1, which is consistent with our previous study [62].

We recognize that this study has certain limitations. The effects of 27-OHC on brain cholesterol metabolism and AD-like pathology were investigated using ApoE ε4 transgenic mice but the specific signal pathways were not explored. In addition, the causal relationship between dysregulated cholesterol metabolism, Aβ1-42 deposition, and nerve cell damage has not been elucidated in this study. Notwithstanding the above limitations, this study also has certain strengths. Taken together, 27-OHC could exacerbate the imbalance of peripheral and brain cholesterol in ApoE ε4 mice and induce Aβ1-42 deposition and synaptic damage, resulting in impaired learning and memory function. This suggests that brain cholesterol homeostasis is an important factor affecting learning and memory abilities. Furthermore, the reversal of abnormal cholesterol metabolism induced by excessive 27-OHC is a promising research direction for the prevention and treatment of AD.

## 4. Materials and Methods

### 4.1. Animals and Treatments

The 12-month-old C57BL/6J mice (SPF, male) were selected from the Laboratory Animal Department of Capital Medical University. ApoE ε4 transgenic mice (SPF; male) were obtained from Jackson laboratory and genotyped using PCR and gel electrophoresis. According to body weight, C57BL/6J and ApoE ε4 transgenic mice were randomly divided into five groups: WT Control group (C57BL/6J mice), Model group (ApoE ε4 transgenic treated with saline), 27-OHC group (ApoE ε4 mice treated with 27-OHC), ANS group (ApoE ε4 mice treated with 27-OHC synthetic enzyme inhibitor anastrozole (ANS)), and 27-OHC+ANS group (ApoE ε4 mice co-treated with 27-OHC and ANS). The dose of 27-OHC (5.5 mg/kg) and anastrozole (0.2 mg/d) was determined as previously described [15,34]. Control groups were injected with an equal volume of saline (0.2 mL/d). All mice received a daily subcutaneous injection for 21 consecutive days. All mice were raised under specific pathogen-free (SPF) conditions of the Laboratory Animal Department of Capital Medical University, with natural lighting, temperature (20 °C–23 °C), humidity (50–55%), and free access to water and food. All animals were fed with basic feed and weighed every two days. After 21 consecutive days of treatment, behavioral experiments were conducted; then, serum and organs were collected and immediately frozen at −80 °C until subsequent assays.

### 4.2. Morris Water Maze

The Morris Water Maze (MWM) system (JX Company, Shanghai, China) was used to evaluate the spatial learning and memory abilities of mice. Briefly, a circular pool (diameter, 120 cm; height, 50 cm) was divided into four quadrants. A circular platform (diameter, 10 cm; height, 38 cm) was submerged 1 cm below the water surface and placed in the center of the southwest quadrant. Before the experiment, the water was dyed with titanium dioxide and kept at a constant temperature (21 ± 1 °C). The experiment included a training phase (days 1–5) and a test phase (day 6). In the training phase, the mice were put into the water from three different quadrants of the circular pool. The time it takes to find the platform from the starting position in training phase was recorded as the escape latency. If a mouse could not find the platform within 90 s, the experimenter would help it reach the platform and stay on it for 15 s; the escape latency was then recorded as 90 s. On the sixth day, the hidden platform was removed and the mice were put into the water from the northeast quadrant. The frequency of target crossing (the number of times crossed the removed platform area at the final day within 90 s), average speed, the time and distance in target quadrant (relative time and distance spent in the quadrant where the platform was placed before), and mean distance to platform were measured and recorded in detail. The shorter the distance to the platform or the higher other four indicators, the better the learning and memory abilities.

### 4.3. Nissl Staining

Nissl staining was used to observe the morphology of brain tissue. First, the paraffin sections were deparaffinized and stained with toluidine blue aqueous solution, preheated at 50 °C for 20 min, then washed with distilled water and 70% alcohol for 1 min, and separated with 95% alcohol until a clear Nissl was observed. Images were observed and collected under the microscope.

### 4.4. Transmission Electron Microscope

The transmission electron microscope was used to observe the synaptic microstructure of the mouse brain tissue. Fresh hippocampal tissue CA1 area was cut into 1 mm × 1 mm × 1 mm pieces and fixed in 2.5% glutaraldehyde for 2 h, rinsed with arsenic acid buffer three times, 1% osmium acid-fixed for 2 h at 4 °C, rinsed with double-distilled water 3 times, and dehydrated with gradient alcohol and embed with epoxy resin; then, uranyl acetate/lead citrate was used for staining when the tissue sample was positioned under the microscope and cut into slice (1 μm thickness), and images were acquired under a JEM-2100 transmission electron microscope (JEOL, Tokyo, Japan).

### 4.5. Immunohistochemistry

To assess the Aβ1-42 deposition in the brains, fresh brain tissue was fixed in paraformaldehyde for 24 h, embedded in paraffin, and sectioned. The paraffin sections were deparaffinized and then placed in citric acid antigen retrieval solution for antigen retrieval, following by being blocked in 3% BSA solution at room temperature for 30 min; then, Aβ1-42 antibody (1:1000; ab126649; Abcam, Cambridge, UK) was added at 4 °C overnight, washed with PBS, and then incubated with the secondary antibody at room temperature for 50 min. After being washed with PBS three times, spin-dried, and stained with fresh DAB solution, the section was rinsed with tap water and incubated using hematoxylin to stain the cell nucleus. Finally, the slides were dehydrated and observed under a microscope.

### 4.6. Biochemical Assays

The levels of serum total cholesterol (TC), low-density lipoprotein cholesterol (LDL-C), and high-density lipoprotein cholesterol (HDL-C) were quantified using an automated biochemical analyzer-Chemray 800 (Rayto, Shenzhen, China).

### 4.7. Enzyme-Linked Immunosorbent Assay

Enzyme-linked immunosorbent assay (ELISA) was performed to determine the levels of serum ApoE and ApoA1. The serum samples were centrifuged at 4 °C and 3500 rpm for 15 min; then, the supernatant was harvested and stored at −80 °C until use. ELISA was performed according to the protocol provided with the kit manual. Then, the absorbance was determined with a spectrophotometer (BioRad, CA, USA) at OD 450 nm and corresponding to standard curves.

### 4.8. Western-Blotting Analysis

Western blot was performed to analyze protein expression. The brain tissue was homogenized in RIPA lysis buffer containing 1% protease inhibitors PMSF and then centrifuged at 12,000 rpm at 4 °C for 10 min; the protein concentration of the supernatants was detected using a BCA assay protein. A total of 20 μg protein was loaded into separate proteins using polyacrylamide gel electrophoresis, and then transferred to PVDF membranes. The membranes were blocked at room temperature for 1 h with 5% non-fat milk in TBST, and then incubated overnight at 4 °C with various primary antibodies. The membranes were then rinsed 3 × 15 min in TBST and incubated in TBST containing secondary antibodies of goat anti-rabbit IgG (1:2000) or goat anti-mouse IgG (1:2000) for 1 h at room temperature. The membrane scanning and protein band intensity were performed and quantified by the Fusion FX imaging system (Vilber Lourmat, France). All bands were normalized based on the model control group and using β-actin as a housekeeping protein. Each experiment was repeated three times. The antibodies used were as follows: Aβ1-42 (1:1000, 14974t, CST, MA, USA), MAP2 (1:1000, ab32454, abcam, Cambridge, UK), GAP43 (1:50,000, ab75843, abcam, Cambridge, UK), SYN (1:20,000, ab32127, abcam, Cambridge, UK), PSD95 (1:1000, ab13552, abcam, Cambridge, UK), SNAP-25 (1:1000, ab109105, abcam, Cambridge, UK), Arc (1:1000, ab51243, abcam, Cambridge, UK), ABCA1 (1:1000, ab66217, abcam, Cambridge, UK), ABCG1 (1:1000, ab52617, abcam, Cambridge, UK), ABCG4 (1:5000, ab101528, abcam, Cambridge, UK), LDLR (1:1000, MABS26, miliipoer), LRP1 (1:20,000, ab92544, abcam, Cambridge, UK), ApoE (1:1000, ab52607, abcam, Cambridge, UK), CYP46A1 (1:1000, SAB2100523, sigma, USA), and ACAT1 (1:1000, ab168342, abcam, Cambridge, UK).

### 4.9. Statistical Analysis

SPSS 22.0 was used for statistical analysis. For continuous data, the normality test was first performed, and the normally distributed data were expressed as mean and standard deviation (mean ± SD). One-way Analysis of Variance (ANOVA) was used for comparison between groups, and the LSD method was further used for post hoc comparisons. Non-normally distributed data were analyzed using Mann–Whitney U test. Repeated measures analysis was used to compare the escape latency and weight changes among groups. All the statistical tests were two-sided, and a significant level was set at *p* < 0.05. 

## Figures and Tables

**Figure 1 ijms-23-11639-f001:**
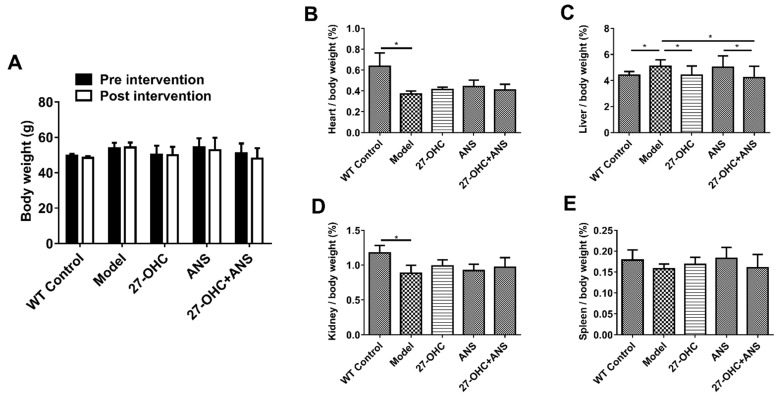
The body weight and coefficients (organ weight/body weight) of mice. (**A**) Body weight of mice pre- and post-intervention. (**B**–**E**) The organ coefficients of heart, liver, kidneys, and spleen (organ weight/body weight, %). All data are shown as mean ± SD. * *p* < 0.05.

**Figure 2 ijms-23-11639-f002:**
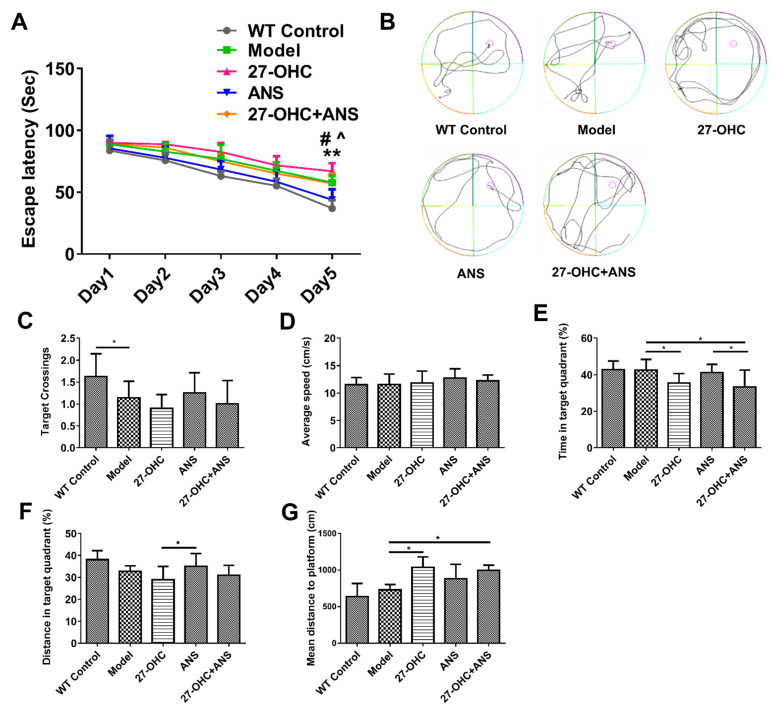
Morris Water Maze test. (**A**) The escape latency of mice changes with time during the training phase. **—All five groups: Day 1 vs. Day 5, *p* < 0.01; **^**—27-OHC group vs Model group, *p* < 0.01; **#**—27-OHC group vs 27-OHC+ANS group, *p* < 0.05. (**B**) The paths of mice in the process of exploring the platform. (**C**) The frequency of target crossing. (**D**) The average speed. (**E**) The time in target quadrant. (**F**) The distance in target quadrant. (**G**) Mean distances to platform were measured by Morris water maze test. All data are shown as mean ± SD (repeated measures ANOVA and one-way ANOVA). * *p* < 0.05; ** *p* < 0.01.

**Figure 3 ijms-23-11639-f003:**
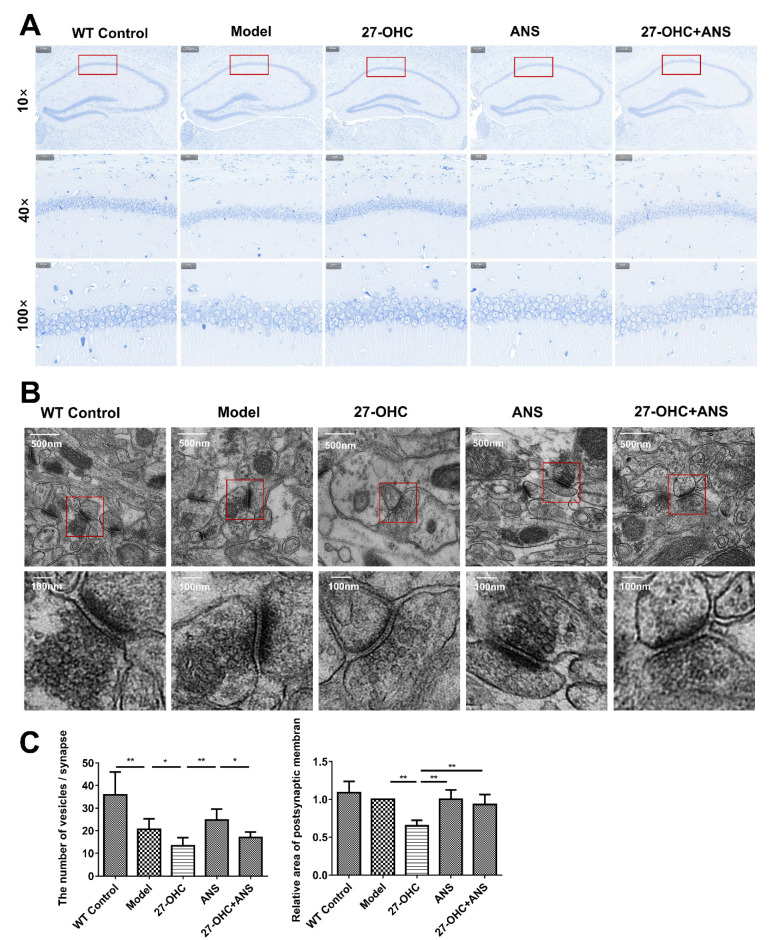
Brain tissue structure and nerve synapses in mice. (**A**) Nissl staining of brain after invention in each group under different magnifications:10× (scale bar = 200 μm), 40× (scale bar = 50 μm), and 100× (scale bar = 20 μm). (**B**) Observation and statistical analysis of the synaptic microstructure of hippocampus CA1 area under a transmission electron microscope. Low magnitude (scale bar = 500 nm); high magnitude (scale bar = 100 nm). (**C**) Statistical analysis of synaptic vesicle and postsynaptic membrane area. All data are shown as mean ± SD. * *p* < 0.05; ** *p* < 0.01 (one-way ANOVA).

**Figure 4 ijms-23-11639-f004:**
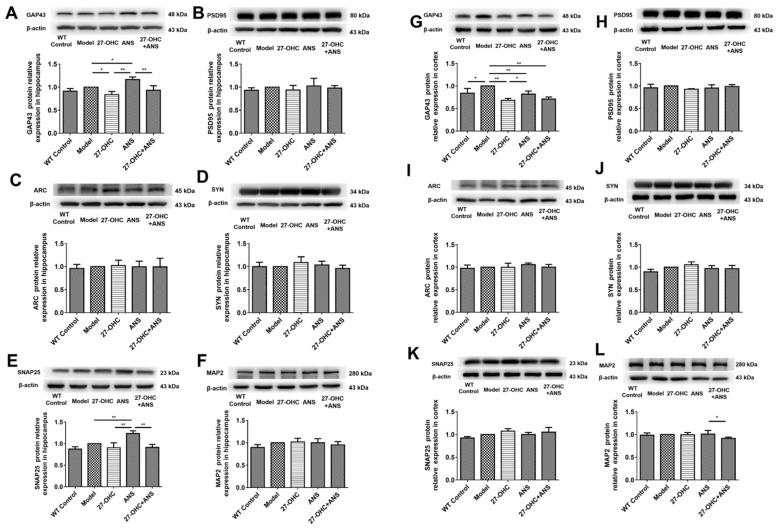
The protein expression of synapse-associated protein in the hippocampus and cortex: (**A**) GAP43 in the hippocampus; (**B**) PSD95 in the hippocampus; (**C**) Arc in the hippocampus; (**D**) SYN in the hippocampus; (**E**) SNAP25 in the hippocampus; (**F**) MAP2 in the hippocampus; (**G**) GAP43 in the cortex; (**H**) PSD95 in the cortex; (**I**) Arc in the cortex; (**J**) SYN in the cortex; (**K**) SNAP-25 in the cortex; (**L**) MAP2 in the cortex; Densitometric analyses were normalized to β-actin. All data are shown as mean ± SD. * *p* < 0.05; ** *p* < 0.01 (one-way ANOVA).

**Figure 5 ijms-23-11639-f005:**
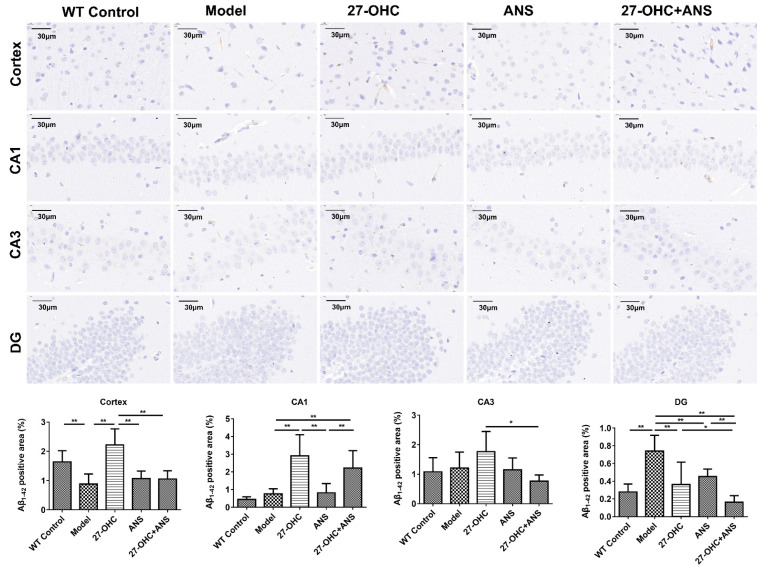
The levels of Aβ1-42 in the cortex and hippocampus. Immunohistochemical was performed to assess the levels of Aβ1-42 in the cortex, hippocampal CA1, CA3, and DG areas (scale bar = 30 μm). The Aβ1-42 positive area was analyzed using Image J. All data are shown as mean ± SD. * *p* < 0.05; ** *p* < 0.01 (one-way ANOVA).

**Figure 6 ijms-23-11639-f006:**
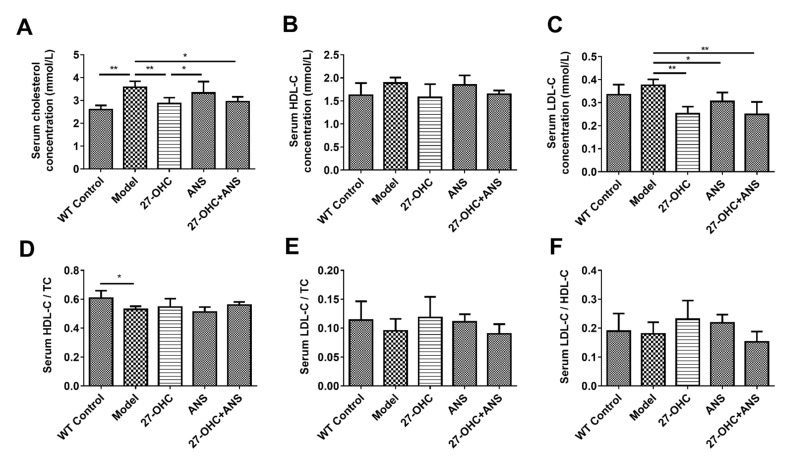
Serum cholesterol levels: The level of (**A**) serum TC; (**B**) serum HDL-C; (**C**) serum LDL-C; (**D**) HDL-C/TC ratio; (**E**) LDL-C/TC ratio; (**F**) LDL-C/HDL-C ratio; (n = 4/group). * *p* < 0.05; ** *p* < 0.01.

**Figure 7 ijms-23-11639-f007:**
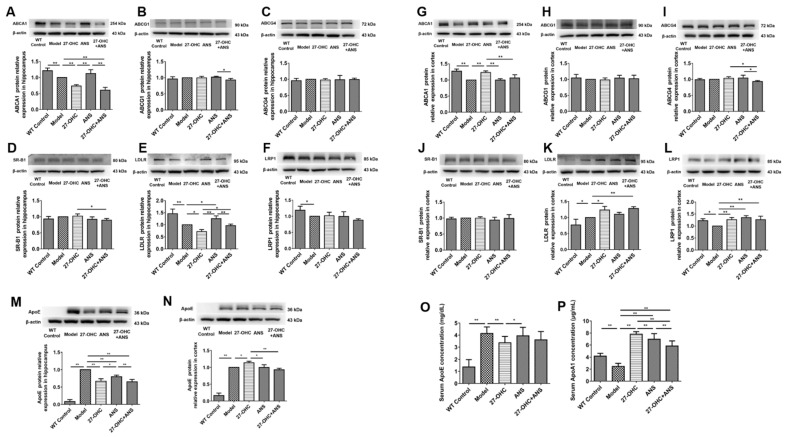
Expression of cholesterol transport-related proteins in the brain and serum: (**A**–**F**) the protein expression in the hippocampus—(**A**) ABCA1 protein, (**B**) ABCG1 protein, (**C**) ABCG4 protein, (**D**) SR-B1 protein, (**E**) LDLR protein, and (**F**) LRP1 protein; (**G**–**L**) the protein expression in the cortex—(**G**) ABCA1 protein, (**H**) ABCG1 protein, (**I**) ABCG4 protein, (**J**) SR-B1 protein, (**K**) LDLR protein, and (**L**) LRP1 protein; (**M**) ApoE protein expression in the hippocampus; (**N**) ApoE protein expression in the cortex; (**O**) Serum ApoE concentration; (**P**) Serum ApoA1 concentration; n = 5/group. Densitometric analyses were normalized to β-actin. All data are shown as mean ± SD. * *p* < 0.05; ** *p* < 0.01.

**Figure 8 ijms-23-11639-f008:**
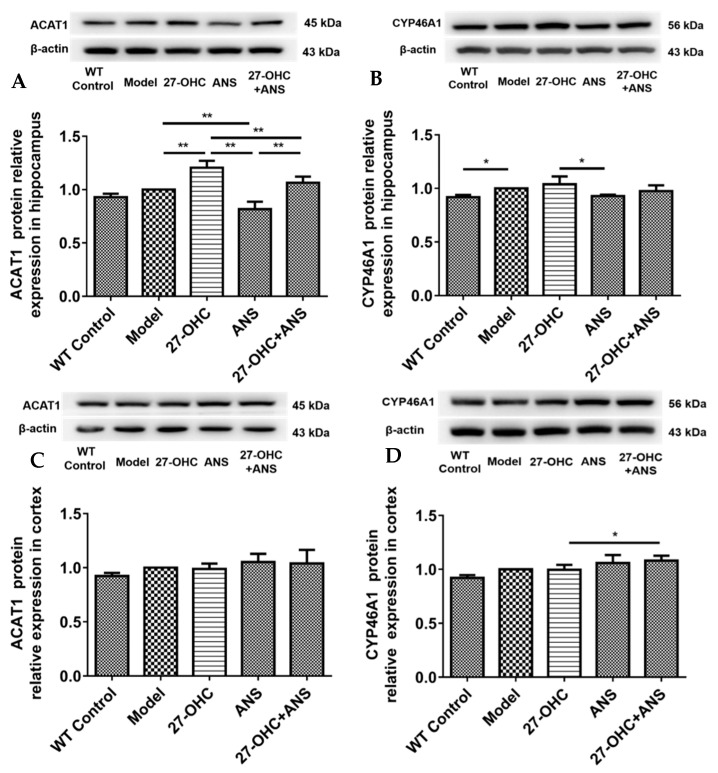
Expression of brain cholesterol conversion-related proteins: (**A**) ACAT1 protein expression in the hippocampus; (**B**) CYP46A1 protein expression in the hippocampus; (**C**) ACAT1 protein expression in the cortex; (**D**) CYP46A1 protein expression in the cortex; n = 5/group. Densitometric analyses were normalized to β-actin. All data are shown as mean ± SD. * *p* < 0.05; ** *p* < 0.01 (one-way ANOVA).

## Data Availability

Data described in the manuscript is available from the corresponding author on reasonable request.

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
