# Peer review of "27-Hydroxycholesterol-Induced Dysregulation of Cholesterol Metabolism Impairs Learning and Memory Ability in ApoE ε4 Transgenic Mice"

_ijms, 2022, doi:10.3390/ijms231911639_

Round 1

Reviewer 1 Report

Manuscript number:       ijms-1837736

Title:                             27-Hydroxycholesterol Impaired Learning and Memory Ability of ApoE ε4 Transgenic Mice by Regulating Cholesterol Metabolism

Recommendation:         Major revision

Comments to authors

The authors in “27-Hydroxycholesterol Impaired Learning and Memory Ability of ApoE ε4 Transgenic Mice by Regulating Cholesterol Metabolism” aimed to prove the hypothesis that 27-Hydroxycholesterol (27-OHC) induced the memory loss, causing the abnormalities in brain tissue, synapse structure, and cholesterol metabolism in ApoE ε4 Transgenic Mice, discussing the possibility that there is a link between 27-OHC and Alzheimer's disease.

The title should be corrected, as it suggested that 27-OHC regulated the cholesterol metabolism, instead of dysregulating the mentioned metabolism.

The Abstract should be rewritten – it is unclearly written (lines16, 17; lines 21-21, lines 21-23, etc…). besides, numerous unexplained abbreviations used in abstract made it difficult to have clear picture what was done, why and what was the outcome of the performed experiments

Please, correct English throughout the text and before submitting consult the English language native speaker (as already given points in Abstract, Introduction, page 1, line: 34, change the “…exhibit…” with more appropriate word; page 1, lines: 39-41; page 1, line: 42, page 2, line: 62 – “…increased…” should be replaced with more appropriate phrase….; page 2, lines: 66,67…….discussion, page 9, lines: 305-311; ……. Materials and methods, page 10, lines: 336,337….etc.)

Introduction:

Page 2, line 66: What kind of derivation did the authors have in a mind?

Results:

Figures 1,2: there is missing explanation of the statistical significance

All figures should be of better quality (they are unclear and it is not possible to distinguish the letters, when not magnified).

Page 4, line 117: please, explain how the synaptic-related proteins were detected in the study.

Discussion:

Page 9, line 319: please rewrite the sentence : “Nonetheless, this study still has some research value.”

Materials and Methods:

Page 10, line 337: please, explain the used doses in the experiments…

Abbreviations are not correctly introduced throughout the text. Please, check and correct all the missing….

What is the difference between apoE, ApoE and APOE, as all figured in the text?

In abstract, in the legends of figures, there is Aβ1-42, in the text there is only Aβ. Please, uniform

Reviewer 2 Report

The authors aim to demonstrate that 27-Hydroxycholesterol impair learning and memory of ApoE ε4 transgenic mice by regulating cholesterol metabolism. To this end they used ApoE ε4 transgenic mice, treated with 27-OHC or without 27-OHC 16 synthetase (CYP27A1) inhibitor anastrozole (ANS) and C57BL/6J mice used as controls.

The theme is interesting and experiments are conducted in a sufficiently good way but methods and above all results are not well described. It is not clear, based on described results, which are the main endpoints of the presented work. Results and figures are also not well described and it is difficult to follow the thread of authors’ descriptions.  So, I suggest to completely rewrite the manuscript prior to resubmit it in a more comprehensive way.

Major points

1) English language must be thoroughly revised by an English native speaker; several sentences are poorly expressed and there are several grammar errors. As an example (line 57):” (CYP27A1) could catalyzes the hydroxylation” must be rewritten as ”(CYP27A1) could catalyze the hydroxylation”…….or (lines 60-63): “In addition, our previous studies have shown that 27-OHC contributes to cognitive decline in APP/PS1 and C57BL/6J mouse[30], increased Aβ in vitro and in vivo models[30, 31], and damaged synaptic structural and functional in primary hippocampal neurons[32]”: contributes, increased and damaged are differently conjugated……or (line 65) what does it mean “motivate the transfer of cholesterol derived in astrocytes to neurons”. Again, what do the authors mean with the sentence “However, whether 27-OHC can interact with APOE regulate cholesterol transport and clearance, then affect cognitive function remains unclear”????? Again in materials and methods section, the sentence “ApoE ε4 transgenic mice (SPF, male) ofthe same age were obtained by breeding ApoEtm1.1 (APOE*4) Adiuj mice purchased from Jackson laboratory and used PCR and gel electrophoresis to identify the genotype of the offspring mice” is not clear

But there are many many unclear sentences throughout the text

Introduction:

Lines 33-37: authors say: “The known driven factors for AD are mutations in the amyloid precursor, presenilin[2], and ApoE genes[3]” Mutations in amyloid precursor and presenilin genes are estimated to account for 1% or less of AD cases, mainly cases with early onset of the disease, being most cases sporadic with no genetic cause (at least not known genetic cause). Further, the APOE is considered a risk gene, not a causative gene in case of mutation, being the E4 allele able only to increase the likelihood of developing AD. Please specify this point also here (not only as the authors do in lines 46-48). Further, the next part of the sentence: “…and exhibit amyloid plaques and neurofibrillary tangles pathology in the brain[4], resulting in atrophy of the cerebral cortex and hippocampus, nerve cell structure and function damage, alteration of energy metabolism and cholesterol metabolism[5-9]” should be separated, as an example with a sentence like: “AD patients exhibit …….”

Lines 39-42: the sentence “Increasing evidence has discovered disturbances of cholesterol homeostasis in AD patients and mouse models[10-13], genome-wide association studies showed that the majority of the genes mapped in late-onset AD are known to be involved in cholesterol metabolism pathways[14]” is not clear and should be split and re-written.

Lines 48-50: the sentence “ApoE in astrocytic is closely related to cognitive function in mouse brain, accelerating Aβ aggregation[19, 20], impairing synaptic plasticity and integrity[21], along with breakdown of blood-brain barrier (BBB) [22]” is not clear, do the authors mean APOE4 allele and astrocytes? Please modify.

Lines 52-54: “The above findings suggest that ApoE has an obvious effect on cholesterol metabolism, and may induce multiple pathological changes in AD by regulating cholesterol transport indirectly”: why indirectly?

Line 69: I suggest not to say “In conclusion

Lines 71-72: “the ApoE ε4 transgenic mice, which regulate cholesterol metabolism and simulate the pathology of AD at the genetic level”: the ApoE ε4 transgenic mice does not simulate AD but may be used to study APOE4 effect on AD mice models with familial AD mutations in the presence of APOE4 allele. In this study, the APOE4 allele only effect has been studied, independently of familial AD classic mutations. Please, rewrite this sentence.

I also suggest a brief explanation of cholesterol metabolism, how, where and when 27-OHC is produced in the body. This is central to better understand the authors’ work.

Results

Lines 82-85: “The heart (F=18.312, P<0.001) and kidney (F=10.441, P<0.001) coefficient was decreased in the Model group compared with the WT group, the liver coefficient in the Control group (P=0.035), 27-OHC group (P=0.048) and 27-OHC +ANS group (P=0.015) was decreased compared with the Model group (Fig. 1-B).” It is not clear what is the so-called “Model group” (APOE4 mice???). In some points this group is named Model group, in other points APOE4 control. This is confounding. Please, specify for a better comprehension of readers. In figure 1B, I see that heart coefficient is decreased (it seems significantly) in all models compared with WT control mice, not only for APOE4 (model??) mice: why statistical significance is highlighted only between wt control mice and APOE (E4?) model??? Please also deeply comment on liver coefficient. Please also comment on kidney (significant only for APOE4 model?????) and spleen coefficient, otherwise do not show those data if you do not think they are central for the results.    

2.2. Morris Water Maze: see also comments on Materials and Methods. What do authors mean with escape latency? Please specify. What do authors mean with F=116.854??? Please specify here and in Methods section.

On day 6, the proportion of time in the target quadrant of mice was significantly decreased in 27-OHC group, and the mean distance to the platform of mice was increased in the 27-OHC group and 27-OHC+ANS group compared with the Model group, which demonstrated that the memory and learning functions were greatly impaired in mice with 27-OHC treatment” please refer data to the figure (2A, 2C????). Please, better define each group of mice (27-OHC group are APOE4 mice treated with 27-OH etc). Also, in 27-OHC+ANS group ANS treatment does not improve memory and learning functions, isn’t it? Please comment on.

Further, Figure 2C describes frequency of target crossing, average speed, time, and distance in the target quadrant but authors do not comment on these measures? Why????? Also, all recorded data are not described in methods section, please describe in depth.

2.3. 27-OHC damages brain tissue structure and nerve synapses in mice:” The hippocampus of WT mice has a clear structure, and the vertebral body layer cells can be seen to be intact, layered, and arranged tightly. In the Model group, 27-OHC group, ANS group, and 27-OHC+ANS group, some cells had incomplete morphology, unclearnucleoli, decreased number of Nissl bodies, and certain cells showed vacuole-like changes (Fig. 3-A).” I sincerely do not see differences in Fig. 3A, the image definition does not allow authors’ conclusions to be drawn.

To further observe the effect of 27-OHC on the plasticity of nerve synapses, we observed the microstructure of synapses under a transmission electron microscope and detected the expression of related proteins. As shown in figure 3-B, after 27-OHC treatment, the area of synaptic vesicles (P=0.01) and postsynaptic membrane (P<0.001) in the hippocampal CA1 region was significantly reduced.” I sincerely do not see differences in Fig. 3B, the image definition does not allow authors’ conclusions to be drawn.

Figure 3C and 3D are not clear, characters are completely unreadable so what the authors describe in the text is impossible to review and comment. Further, authors should explain why they used markers like GAP43 etc to demonstrate synapse damage (not all readers know established synapse damage markers…….) “Synaptic-related proteins were also detected in our study. In the hippocampus, compared with the Model group, the expression of GAP43 in the 27-OHC group (P=0.012) was down-regulated, GAP43 (P=0.011) and SNAP25 (P=0.002) in the ANS group was significantly up-regulated (Fig. 3-C). In the cortex area, compared with the Model group, the expression of GAP43 in the 27-OHC group (P<0.001), ANS group (P=0.006), and 27-OHC+ANS group (P<0.001) was down-regulated, and the expression of GAP43 in the ANS group was significantly up-regulated (Fig. 3-D)

2.4. 27-OHC increases Aβ burden in hippocampus and cortex of ApoE4 mice

To detect the Aβ burden in the brain, immunohistochemical were used, and the results showed that there were fewer Aβ-positive areas in each group, but the proportion of the cortical area (P<0.001) and hippocampal CA1 area (P<0.001) in the 27-OHC group was significantly higher than that in the Model group (Fig. 4)” Please comment on DG area and significant differences

2.5. Brain and serum cholesterol levels 144

As shown in figure 5, compared with the Model group, the HDL-C was decreased after 27-OHC treatment in the cortex (P<0.001). The LDL-C was increased after 27-OHC treatment in both hippocampus (P=0.002) and cortex (P<0.001) (Fig. 5-A, B). In the serum, compared with the Model group, the level of total cholesterol (TC) was decreased in the 27-OHC group (P=0.008) and 27-OHC+ANS group (P=0.015). The level of LDL-C was decreased in the 27-OHC group (P=0.003), ANS group (P=0.035) and 27-OHC+ANS group (P=0.001).

Characters are unreadable, importantly authors do not comment on significant differences between all groups. It is difficult in this way to understand what the end points of this research work are. Differences between wt mice and apoe4 mice or apoe4 mice with or without 27-OHC treatment +/- ANS are not well described and commented.

Figure 6: characters are unreadable, so it is difficult to comment on

Materials and methods

Line 332: “According to body weight, C57BL/6J and APOE mice were randomly divided into 5 groups”. Maybe, age is important, not only body weight.

Line 337: “27-OHC (5.5mg/kg) and anastrozole (0.2mg/day)……” Please deeply explain why those doses were used. This is a central theme.

4.2. Morris Water Maze. Please better explain how the tests are made. Explanations are not consistent with what the authors show in Figure 2. What do authors mean with escape latency? Please specify. Figure 2C describes frequency of target crossing, average speed, time, and distance in the target quadrant but authors do not explain these measures. Please specify.

Reviewer 3 Report

The aim of the current study is very interesting but the authors need to describe more extensively their results and analyses further their data to support their findings. Specifically:

1) Figure 2C: The results presented in each panel of Figure 2C should be described in detail in text. Each panel should be numbered separately. It seems there are more statistically significant differences between treatments than those that are reported.  

2) Figure 3: the  same applies to Figure 3C. Furthermore, the whole Figure 3 should be enlarged. The authors should consider splitting figure 3 to more than one figure.

3) HDL and LDL exist in plasma, not in brain. What kit did you use?  It is not clear what you measure in figure 5A and B.

4) Figure 6: Panels A-C should be enlarged. The results presented in each panel of should be described in detail in text. It seems there are more statistically significant differences between treatments than those that are reported. 

Reviewer 4 Report

This is a very nicely written and discussed manuscript which presents some interesting findings related to cholesterol. 

My only minor comment relates to the figures (3-6) which are too small. It is impossible to appreciate the images as well as read the text within the graphs when the figures are that compressed. I suggest that the authors reorganize these. 

Also, in figure 7 the authors used beta actin (43kDa) as loading control for a protein, ACAT1 (45kDa) of similar size. Was this done on separate gels or is this in fact a true loading control? This needs to be clarified. 

Round 2

Reviewer 1 Report

Manuscript number:       ijms-1837736

Title:                             27-Hydroxycholesterol-Induced Dysregulation of Cholesterol Metabolism Impaired Learning and Memory Ability in ApoE ε4 Transgenic Mice

Recommendation:         Minor revision

Comments to authors

The authors improved their manuscript.

Still, English should be checked throughout the text.

Reviewer 2 Report

Although authors sufficiently replied to most comments, I think again that an extensive English language editing is needed prior to publication

Reviewer 3 Report

The authors measure HDL-C and LDL-C in brain. Mouse has minimal, mostly absent, levels of LDL in plasma, so it is not clear what the authors measure in mouse brain as LDL-C. I'm also not convinced for the presence of apoA-I-HDL in high levels in mouse brain. Has apoA-I-HDL detected in high levels in mouse brain in other studies? 

Round 3

Reviewer 3 Report

This is the 3rd review of the manuscript. At the original submission and subsequent revision the authors had reported the measurement of HDL-C and LDL-C in brain. Lipoproteins in brain are different than those in periphery. The authors should read the excellent review paper “Lipoproteins in the Central Nervous System: From Biology to Pathobiology. Annu. Rev. Biochem. 2022. 91:14.1–14.29”. In the current version of the manuscript the reported findings on brain “HDL-C and LDL-C” have been removed.

Although I do appreciate that the authors recognized that my comments were correct and followed my suggestions, I still think that the authors lack a thorough understanding of lipoproteins and lipid transport in brain and the appropriate measurements that can be used. Therefore, I won’t endorse this study for publication in the prestigious journal IJMS.
